# Hand-Object Interaction Image Generation

**Hezhen Hu**[1]    **Weilun Wang** [1*]   **Wengang Zhou**[1,2†]    **Houqiang Li**[1,2†]

[1]CAS Key Laboratory of GIPAS, EEIS Department
University of Science and Technology of China (USTC)
[2] Institute of Artificial Intelligence, Hefei Comprehensive National Science Center
{alexhu, wwlustc}@mail.ustc.edu.cn
{zhwg, lihq}@ustc.edu.cn

## Abstract

In this work, we are dedicated to a new task, *i.e.,* hand-object interaction image generation, which aims to conditionally generate the hand-object image under the given hand, object and their interaction status. This task is challenging and research-worthy in many potential application scenarios, such as AR/VR games and online shopping, *etc.* To address this problem, we propose a novel HOGAN framework, which utilizes the expressive model-aware hand-object representation and leverages its inherent topology to build the unified surface space. In this space, we explicitly consider the complex self- and mutual occlusion during interaction. During final image synthesis, we consider different characteristics of hand and object and generate the target image in a split-and-combine manner. For evaluation, we build a comprehensive protocol to access both the fidelity and structure preservation of the generated image. Extensive experiments on two large-scale datasets, *i.e.,* HO3Dv3 and DexYCB, demonstrate the effectiveness and superiority of our framework both quantitatively and qualitatively. The code will be available at https://github.com/play-with-HOI-generation/HOIG.

## 1 Introduction

As a crucial step for analyzing human actions, hand-object interaction understanding is research-worthy in a broad range of applications related to virtual or augmented reality. Current works largely focus on hand-object pose estimation (HOPE) [16, 19, 21], which aims to capture the pose configuration of the given hand-object image. In contrast, its inverse counterpart is seldom considered. In this work, we aim to explore this novel task and term it Hand-Object Interaction image Generation (HOIG). Its objective is to generate the interacting hand-object image under the guidance of the target posture, while preserving the appearance of the source image, as illustrated in Figure 1.

This HOIG task is of both application and research value to the community. On one hand, HOIG can be potentially applied to many scenarios, such as AR/VR games, online shopping, and data augmentation, *etc.* For online shopping, HOIG can give the consumer an immersive experience and flexibility on object customization. On the other hand, HOIG is of board research interest. Since it explicitly involves simultaneous generation of two instances (hand and object) with high interaction relationship, HOIG brings many new challenges to resolve. These characteristics thus bring new challenges. 1) It requires to process the complex self- and mutual occlusion between the interacting hand and object. 2) It involves image translation of co-occurring instances, where different appearance characteristics of the hand and object need to be considered.

---

*Contribute equally with the first author.
†Corresponding authors: Wengang Zhou and Houqiang Li.

36th Conference on Neural Information Processing Systems (NeurIPS 2022).

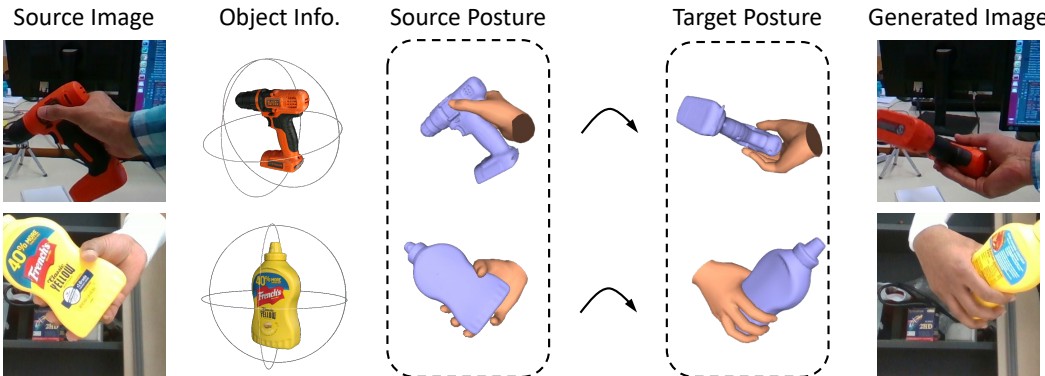

| Source Image | Object Info. | Source Posture | | Target Posture | Generated Image |

Figure 1: Definition of our proposed task. It aims to generate the target interacting hand-object image under the target pose, while persevering the source image appearance.

The effectiveness of Generative Adversarial Networks (GANs) has been validated in realistic image synthesis, such as human, face and hand [8, 27, 29, 39]. These GAN-based synthesis methods can be conditioned on different input information, such as simple-to-draw sketches, 2D sparse keypoints and dense semantic masks, *etc.* Among them, GestureGAN [39] resolves the isolated hand generation, which is mostly related to our task. It utilizes 2D sparse hand keypoints as the condition and attempts to generate the target hand image according to the optical flow learned from the source and target conditions. However, these methods do not consider the generation of two interacting instances, which cannot meet the requirement of the HOIG task.

In this work, we propose the HOGAN framework to deal with the challenges of this novel task. HOGAN utilizes the expressive model-aware representation as the condition and leverages its inherent structure topology to build the unified hand-object surface space. In this space, complex self- and mutual occlusion among hand and object are explicitly modeled. Specifically, the visible parts of hand and object are mapped to the target image plane, along with their corresponding fine-grained topology map. Meanwhile, the transformation flow is computed between the source and target. These middle results provide abundant information for final image synthesis. During synthesis, our framework considers appearance difference between hand and object and generates the target hand-object image in a split-and-combine manner.

To systematically explore this task, we build the comparison baselines via adopting representative methods from the most related single hand generation task with a few modifications. The image generation quality is carefully evaluated from multiple perspectives, including the fidelity (FID and LPIPS), structure preservation (AUC, PA-MPJPE and ADD-0.1D) and subjective user study.

Our contributions are summarized as follows,

- To our best knowledge, we are the *first* to explore the novel task named the hand-object interaction image generation, *i.e.,* conditionally generating the hand-object image under the given hand, object and their interaction status. This task is of board research and application value to the community.
- To deal with the challenges of this task, we propose the HOGAN framework, which considers the hand-object occlusion and generates the target image in a split-and-combine manner.
- To systematically explore this task, we present comparison baselines from related single-hand generation. Besides, the comprehensive metrics are chosen to evaluate the both fidelity and structure preservation of the generated image. Extensive experiments on two datasets demonstrate the effectiveness and superiority of our method over baselines.

## 2   Related Work

In this section, we will briefly review the related topics, including pose-guided image synthesis and hand-object interaction.

**Pose-guided image synthesis.** Pose-guided image synthesis is a conditional generation task, which aims to generate an image under the condition of the target pose while preserving the identity of

the source image. This problem involves instances with rigid parts such as bodies [1, 5, 11, 12, 13, 20, 29, 36, 34, 38, 45], faces [8, 18, 24, 28, 35], and hands [17, 22, 39, 40], and can be utilized in various scenarios, such as image animation, face reenactment, and sign language production, *etc*. In recent work, Ren *et al.* [29] generate person images under target posture with a differential global-flow local-attention framework in a multi-scale manner. Deng *et al.* [8] utilize 3DMM [2] which parameterizes pose and shape to disentangle posture representation for face generation. Hu *et al.* [17] incorporate hand prior for pose-guided hand image synthesis instead of 2D joint representation. However, previous methods mainly focus on single-object pose translation problems. Different from them, the novel problem we formulate, *i.e.*, HOIG, involves generation of the co-occurring subjects, which brings new characteristics worth exploring.

**Hand-object interaction.** Most current works on hand-object interaction focus on simultaneously estimating hand-object pose aligning the given image [4, 7, 9, 15, 16, 19, 21, 23, 25, 32, 42]. To better depict hand-object interaction, they resort to dense triangle meshes with the pre-defined topology as the representations, which are produced by the MANO hand model [30] and the known object model [3, 41]. Hasson *et al.* [16] leverage physical constraints to better estimate hand and object meshes. Cao *et al.* [4] propose a optimization-based method, which leverages 2D image cues and 3D contact priors for hand-object interaction estimation. Liu *et al.* [21] further boost the estimation performance by semi-supervised learning with the assistance of in-the-wild hand-object videos. To our best knowledge, there exists no work on the inverse task of hand-object interaction estimation. In this work, we aim to explore this novel task and term it hand-object interaction image generation.

## 3 Methodology

In this section, we first discuss our problem formulation. Then we elaborate the architecture of our proposed HOGAN framework. Finally, we introduce the loss function during optimization.

### 3.1 Problem Formulation

Hand-object interaction image generation is a conditional generation task. Its objective is to generate the interacting hand-object image under the target pose condition, while preserving the source appearance. Specifically, the object is well-modeled with texture known. Resolving this task is challenging, since it is non-trivial to understand the complex interacting relationship between hand and object during image generation.

We summarize the main challenges as follows. *Firstly*, it requires modeling the occlusion between co-occurring instances. In the interacting hand-object scenario, complex self- and mutual occlusion usually occur. Since the occlusion leads to more transformation complexity between the source and target, the occluded regions should be located and identified, which eases the final synthesis. *Secondly*, it needs to take into account different characteristics of two instances during generation. Specifically, hand is articulated and encounters self-occlusion among joints, while object is usually rigid with fine-grained texture. The framework should generate realistic appearance of co-occurring instances, jointly with reasonable manipulation between them.

### 3.2 HOGAN

**Overview.** As illustrated in Figure 2, our framework first utilizes the expressive model-aware representation as the pose condition. Then we leverage the pre-defined structure in the model to build the unified space to perform occlusion-aware topology modeling. This modeling identifies the occluded regions, and provides the coarse target images and fine-grained topology maps for the next stage. Finally, hand-object synthesis considers the appearance difference of these instances, and generates the target hand-object image in a split-and-combine manner.

**Occlusion-aware topology modeling.** We first give an overview of the utilized model-aware representation. The hand and object are represented via MANO [30] and YCB [3, 41] models, respectively. Both MANO and YCB provide triangulated meshes, which can densely depict the structure of hand and object. Specifically, the hand-object joint representation contains $N_v$ vertices and $N_f$ faces. Its mesh $\mathbf{H} \in \mathbb{R}^{N_v \times 3}$ represents the vertex coordinates. The inherent topology $\mathbf{P} \in \mathbb{R}^{N_f \times 3 \times 2}$ is organized as a vertex triplet, in which each unit is recorded as corresponding vertex

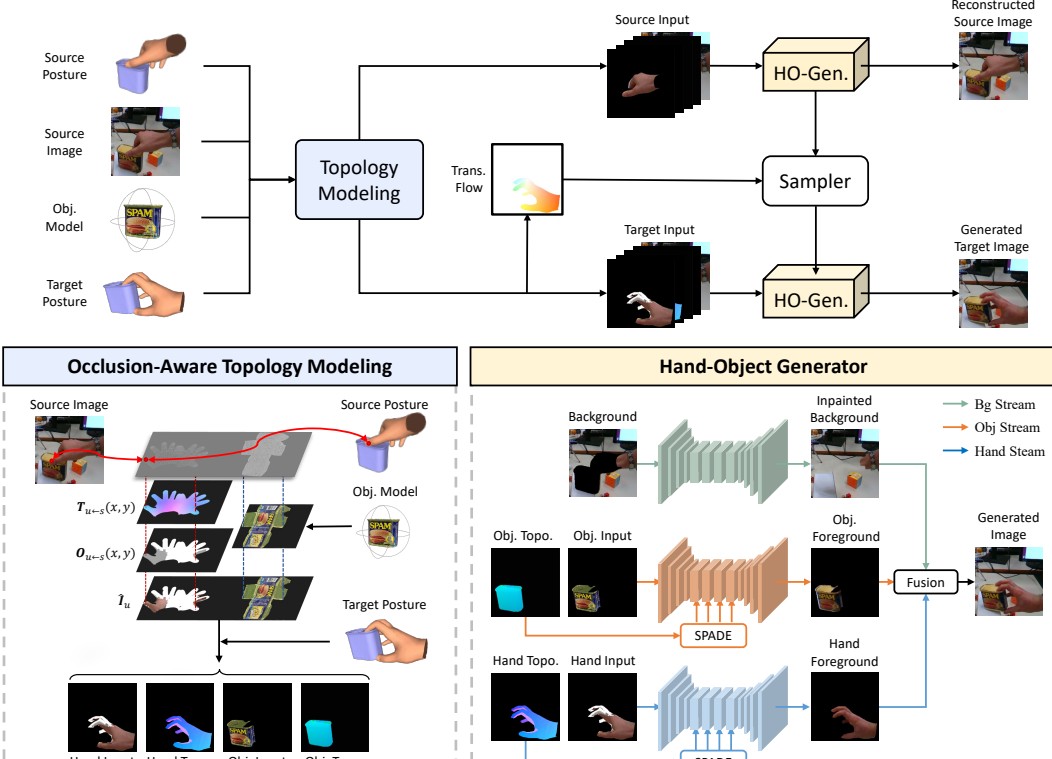

Figure 2: Overview of our proposed HOGAN framework. It leverages the expressive model-aware representation as the pose condition, jointly with its inherent topology to build the unified surface space. Embedded with this space, we explicitly model complex self- and mutual occlusion and generate the coarse image and fine-grained topology map for the next stage. Hand-object generator takes into account the different characteristics between these two instances and produces the final target image.

coordinates aligning the desired plane. $s$, $t$ and $u$ are notations for the source, target and unified surface space, respectively.

Embedded with the pre-defined topology, we unravel the surface of hand and object model to build the unified space. In this space, the same representation is binded with the same mesh face, ignoring the pose configuration. This space is adopted to perform mapping from the source to the target, while inserting the pre-known object texture. Firstly, we leverage the source pose to map its corresponding image appearance to this unified space in an occlusion-aware manner as follows,

$$\mathbf{T}_{u\leftarrow s}(x,y) = \mathbf{W}^u(x,y) \cdot \mathbf{P}^s(\mathbf{F}^u(x,y)), \tag{1}$$

where $\mathbf{T}_{u\leftarrow s}(x,y)$ denotes the flow from the source image to the unified space. $\mathbf{F}^u(x,y)$ denotes the face index belonging to the $(x,y)$ location in the surface space, and $\mathbf{W}^u(x,y)$ represents the relative weighted position in this face. Besides, occlusion should be jointly computed to locate the visible texture as follows,

$$\mathbf{O}_{u\leftarrow s}(x,y) = (\mathbf{F}^u(x,y) \neq \mathbf{F}^s(\mathbf{T}_{u\leftarrow s}(x,y))). \tag{2}$$

The visible texture from the source image is mapped to our unified surface space as follows,

$$\mathbf{I}_u = \mathrm{Warp}(\mathbf{T}_{u\leftarrow s}, \mathbf{I}_s) \odot \mathbf{O}_{u\leftarrow s}, \tag{3}$$

where $\mathbf{I}_u$ aligns our surface space. $\odot$ and $\mathrm{Warp}(\cdot)$ represents the element-wise multiplication and warping function, respectively. Notably, the object in the source image inevitably contains the occluded region, which is insufficient for the target generation. Therefore, we utilize the pre-stored object texture to replace the original object region in $\mathbf{I}_u$, resulting new texture image $\hat{\mathbf{I}}_u$. Finally, we compute the flow performing the mapping between the unified space to the target image.

$$\mathbf{T}_{t\leftarrow u}(x,y) = \mathbf{W}^t(x,y) \cdot \mathbf{P}^u(\mathbf{F}^t(x,y)), \tag{4}$$

After that, the target image $\mathbf{I}_t$ is generated via sampling $\hat{\mathbf{I}}_u$ under the guidance of $\mathbf{T}_{t \leftarrow u}$ as follows,

$$\mathbf{I}_t = \text{Warp}(\mathbf{T}_{t \leftarrow u}, \hat{\mathbf{I}}_u), \tag{5}$$

$\mathbf{I}_t$ represents the coarse target hand-object image. Meanwhile, to provide more guidance for the next stage, the fine-grained topology map $\mathbf{Y}_t$ is concurrently generated as follows.

$$\mathbf{Y}_t(x, y) = \text{Bary}(\mathbf{P}^u(\mathbf{F}^t(x, y))), \tag{6}$$

where $\text{Bary}(\cdot)$ computes the barycenter of the corresponding face in the surface space.

**Hand-object synthesis.** Considering hand and object exhibit different properties, we design the hand-object generator for target image synthesis in a split-and-combine manner. As illustrated in Figure 2, the hand-object generator consists of three streams: *Background Stream*, *Object Stream* and *Hand Stream*.

The *Background Stream* inpaints the background cropped from the source image with the hand-object foreground mask. The *Object Stream* takes the rendered object images as input and transfers its style to the real image domain. To make the stream aware of object structure information, we adopt the spatially-adaptive normalization (SPADE) [26] to insert the object topology map.

The *Hand Stream* deals with the hand part that is not occluded by the object. We first synthesize a coarse image of the visible part in the target posture by warping the hand in the source image with the pose transformation flow. After that, the coarse image is refined by the U-structure network [31] in the hand stream. During refinement, similar to the *Object Stream*, the hand topology map is modulated into the network with SPADE. Meanwhile, we extract the multi-scale features from the hand-object generator that reconstructs the source image, and integrates them into the generation process with the attention sampler [29].

The three streams separately process three instances with different properties, *i.e.*, background, object and hand, and merge their results with the fusion module. With the features from the last layer of three streams, our framework further utilizes two convolutional layers to learn two fusion masks, *i.e.*, the hand mask $\mathbf{M}_h$ and the hand-object mask $\mathbf{M}_f$, which indicate the area belonging to the unoccluded hand the hand-object foreground, respectively. Regarding these two mask, the fusion module merges the the three-stream results to the final generated results $\mathbf{I}$ as follows,

$$\mathbf{I} = (\mathbf{I}_h \odot \mathbf{M}_h + \mathbf{I}_o \odot (1 - \mathbf{M}_h)) \odot \mathbf{M}_f + \mathbf{I}_b \odot (1 - \mathbf{M}_f), \tag{7}$$

where $\mathbf{I}_b, \mathbf{I}_o, \mathbf{I}_h$ are the generated results of the background, object and hand stream, respectively. $\odot$ refers to the element-wise multiplication.

## 3.3 Objective Functions

We design a discriminator to train our HOGAN in an adversarial learning manner. The adversarial loss constrains the distribution of the generated images with that of the real images, which improves the visual performance of generated images. With the discriminator $D(\cdot)$, the adversarial loss is formulated as follows,

$$\mathcal{L}_{\text{adv}}^G = \mathbb{E}_{\boldsymbol{x}_f, \boldsymbol{c}}[(1 - D(\boldsymbol{x}_f | \boldsymbol{c}))^2],$$
$$\mathcal{L}_{\text{adv}}^D = \mathbb{E}_{\boldsymbol{x}_r, \boldsymbol{c}}[(1 - D(\boldsymbol{x}_r | \boldsymbol{c}))^2] + \mathbb{E}_{\boldsymbol{x}_f}[(1 + D(\boldsymbol{x}_f | \boldsymbol{c}))^2], \tag{8}$$

where the $\boldsymbol{x}_f$ and $\boldsymbol{x}_r$ are the distribution of the generated and real images, respectively. $\boldsymbol{c}$ refers to the combination of target hand and object posture information.

Besides the adversarial loss, we regularize our HOGAN with the reconstruction loss on the source image and the perceptual loss on the generated image, which is formulated as follows,

$$\mathcal{L}_{\text{rec}} = \|\boldsymbol{x}_s - \hat{\boldsymbol{x}}_s\|_1,$$
$$\mathcal{L}_{\text{vgg}} = \sum_i \|f_i(\boldsymbol{x}_t) - f_i(\hat{\boldsymbol{x}}_t)\|_1, \tag{9}$$

where $\boldsymbol{x}_s$ and $\boldsymbol{x}_t$ are the generated source and target images while $\hat{\boldsymbol{x}}_s$ and $\hat{\boldsymbol{x}}_t$ are the ground-truth source and target images, respectively. $f_i(\cdot)$ is the feature extractor from the $i$-th layer of a pre-trained VGG-19 network [37].

The overall loss is the summation of three objective functions as follows,

$$\mathcal{L} = \mathcal{L}_{\text{adv}}^G + \lambda_1 \mathcal{L}_{\text{rec}} + \lambda_2 \mathcal{L}_{\text{vgg}}, \tag{10}$$

where $\lambda_1$ and $\lambda_2$ are the weighting parameters to balance the objective functions.

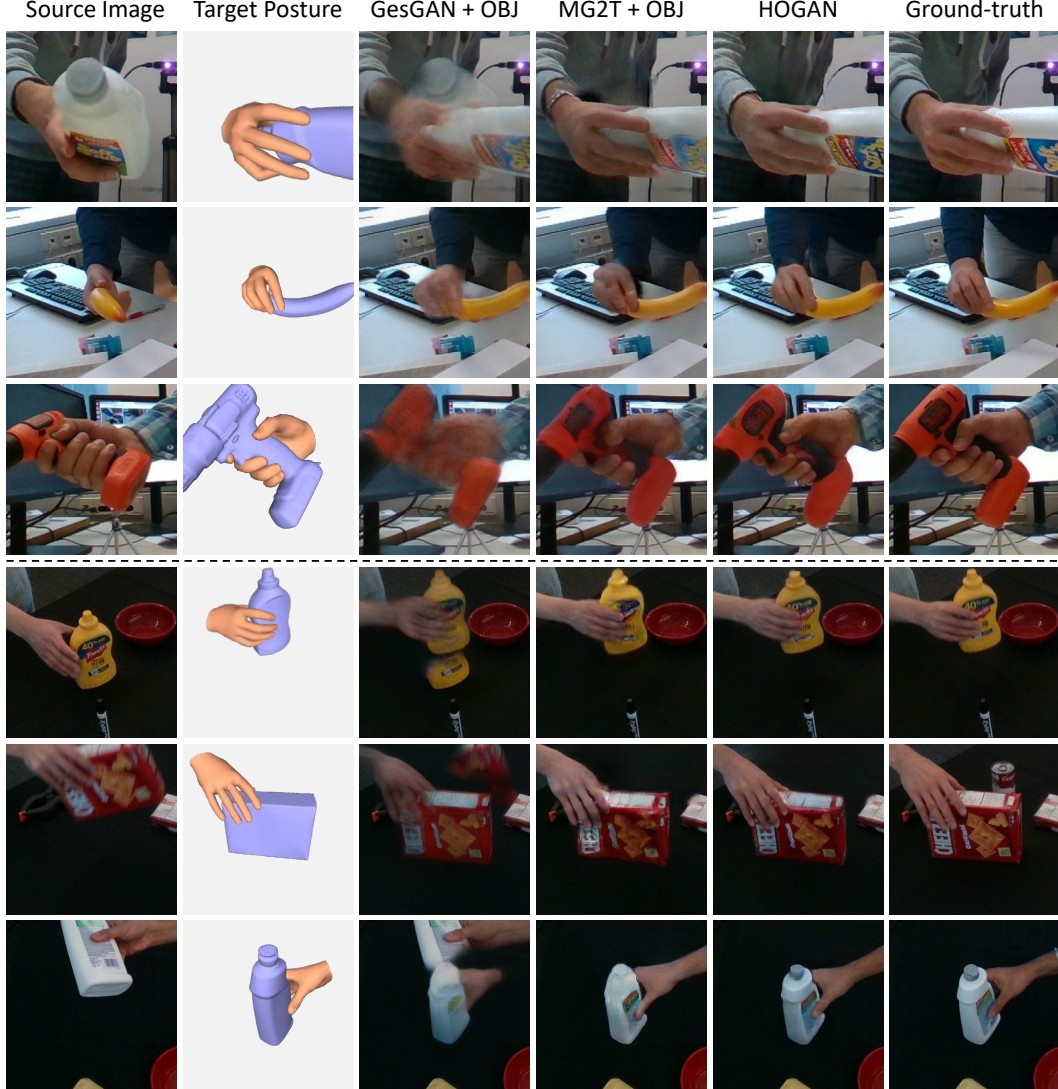

| Source Image | Target Posture | GesGAN + OBJ | MG2T + OBJ | HOGAN | Ground-truth |

Figure 3: Qualitative comparison with baselines, including GestureGAN [39] + OBJ and MG2T [17] + OBJ, on the HO3Dv3 and DexYCB dataset. HOGAN exhibits superior performance, generating images without blurry, texture aliasing and false hand-object interaction.

## 4 Experiment

In this section, we first introduce the experiment setup, including datasets, implementation details, and evaluation metrics. Then we elaborate the baseline methods and make comparison with them both quantitatively and qualitatively. After that, we conduct ablation study to highlight the important components in our framework. Furthermore, we explore more applications of our HOIG task.

### 4.1 Experiment Setup

**Datasets.** We evaluate our method on two large-scale datasets with annotated hand-object mesh representation, *i.e.,* **HO3Dv3** [14] and **DexYCB** [6]. HO3Dv3 is captured in the real-world setting. It contains 10 different subjects performing various fine-grained manipulation on one among 10 objects from YCB models [3]. The training and testing set contain 58,148 and 13,938 images, respectively. DexYCB is recorded in the controlled environment, with 10 subjects manipulating one among 20 objects. In our experiment, we choose the frames containing interaction between hand and object, with 33,562 and 8,554 images for training and testing, respectively.

Table 1: Comparison with two hand-object interaction image generation baselines, *i.e.*, GestureGAN + OBJ and MG2T + OBJ, on the HO3Dv3 and DexYCB dataset. ↑ and ↓ represent the higher the better, and the lower the better, respectively.

| Method | HO3Dv3 | | | DexYCB | | |
|---|---|---|---|---|---|---|
| | FID↓ | LPIPS↓ | UPR↑ | FID↓ | LPIPS↓ | UPR↑ |
| GestureGAN [39] + OBJ | 82.0 | 0.316 | 0.5 | 34.5 | 0.214 | 4.0 |
| MG2T [17] + OBJ | 45.6 | 0.214 | 24.8 | 37.8 | 0.121 | 22.0 |
| HOGAN | **41.3** | **0.171** | **74.7** | **30.1** | **0.109** | **74.0** |

**Implementation details.** The whole framework is implemented on PyTorch and we perform experiments on 4 NVIDIA RTX 3090. All U-Nets are trained from scratch. The hand-object generator is trained in an end-to-end manner and all the networks are being trained simultaneously. The Adam optimizer is adopted and the training lasts 30 epochs. We set the batch size to 8 in our experiment. The learning rate is set as 2e-4 for the first 15 epochs and linearly decays to 2e-6 till the end. The hyperparameter $\lambda_1$ and $\lambda_2$ are set to 10 and 10, respectively.

**Evaluation metrics.** We design an evaluation protocol to measure the hand-object interaction image generation quality both quantitatively and qualitatively. In quantitative evaluation, we measure both the fidelity and structure preservation of generated images. For the image fidelity, we adopt the widely-used Fréchet Inception Distance (FID) [10] and Learned Perceptual Similarity (LPIPS) [44] metrics. To evaluate the posture preservation of hand and object in the generated images, we utilize an off-the-shelf hand-object pose estimator [21] to report AUC and PA-MPJPE for hand pose, and ADD-0.1D for object pose. PA-MPJPE represents the joint mean error after Procrustes alignment, and AUC denotes the area under the 3D PCK curve. ADD-0.1D denotes the percentage of average object vertices error within 10% of object diameter.

In qualitative evaluation, we conduct a user study to evaluate the visual performance of our method and two baselines. There are 20 volunteers participating in this study. In the study, the volunteers are asked to select the most high-fidelity generated results among our method and two baselines. The percent of generated image preferred is recorded as User Preference Ratio (UPR).

## 4.2 Baselines

Since there exists no work on this task, we present two baselines from most related single-hand generation, *i.e.*, GestureGAN [39] and MG2T [17], with a few modifications. These methods are mainly modified with the object condition information involved as follows.

**GestureGAN + OBJ.** GestureGAN translates the single-hand source image to the target posture in a cycle-consistent manner. The source image is fed into the network along with the target posture represented by sparse keypoints. In GestureGAN + OBJ, we concatenate the object rendered result with the target posture and feed them into the network. The modified framework is trained with the same architecture design and objective functions as the previous GestureGAN.

**MG2T + OBJ.** MG2T is also a single-hand generation framework with hand prior incorporated. In MG2T + OBJ, we maintain the hand processing module in MG2T and further extend the object modeling to involve the object posture information. The rendered result of the object is merged into the foreground branch to produce the hand-object translation results.

Rendering-based methods should also be adopted for comparison. Among them, one main procedure is to get the reliable texture. However, the source image only contains the partial hand texture, which leads to the incomplete hand of the target rendered image. Therefore, the direct rendering is not applicable and we resort to MG2T + OBJ, which has incorporated the refinement on rendered results.

## 4.3 Comparison with Baselines

We present quantitative results to evaluate the effectiveness of our method on HO3Dv3 and DexYCB datasets. *Firstly*, we study the fidelity of generated images on two datasets. As shown in Table 1, our method surpasses two baselines on all metrics with a notable gain. *Secondly*, it is essential for hand-object interaction image generation to maintain the posture information of the target hand and object. Therefore, we carefully analyze the posture preservation of generated images. With the

Table 2: Hand-object structure preservation analysis on HO3Dv3 dataset. ↑ and ↓ represent the higher the better, and the lower the better, respectively.

| Method | Hand | | Object (ADD-0.1D) | | | | | |
|---|---|---|---|---|---|---|---|---|
| | AUC↑ | PAJPE↓ | Sugar Box↑ | Bottle↑ | Banana↑ | Mug↑ | Power Drill↑ | Aver.↑ |
| GestureGAN [39] + OBJ | 74.1 | 13.0 | 6.4 | 0.0 | 14.8 | 1.1 | 0.0 | 4.5 |
| MG2T [17] + OBJ | 76.2 | 11.9 | 36.1 | 65.0 | 21.5 | 48.9 | 10.9 | 36.5 |
| HOGAN | **76.5** | **11.8** | **52.1** | **85.0** | **26.3** | **55.0** | **34.1** | **50.5** |
| GT | 76.9 | 11.6 | 76.1 | 100.0 | 30.2 | 64.4 | 49.1 | 63.9 |

off-the-shelf network [21], we estimate the hand and object pose in the generated images and compute the error with the target pose annotation. As shown in Table 2, it can be observed that our method exhibits superior performance for hand and objects of different categories over two other baselines.

Furthermore, we perform qualitative comparisons with baselines. From Figure 3, we observe that the images generated by our method exhibit better visual performance compared with previous methods. Under the complex scenes, where the hand and object are highly interacting, our method can generate images with more reasonable spatial relationship, which significantly outperforms baseline methods. We also conduct a user study to evaluate subjective visual performance. The voting results are reported as the USP metric in Table 1. It is observed that our method is preferred by over 70 percent against two competitors.

### 4.4 Ablation Studies

We perform ablative experiments to highlight several important components, *i.e.*, the hand and object topology, source feature transfer and split-and-combine generation.

**Hand and object topology.** In HOGAN, we modulate both hand and object topology into the generator to provide detailed geometric information of hand and object. In Table 3, we verify its effectiveness by comparing it with non-topology variants. It can be observed that HOGAN (the first row) achieves the best performance over two other variants. Without either hand or object topology, the framework suffers performance degradation, *e.g.* 6.0 and 6.5 on FID, 0.012 and 0.015 on LPIPS.

Table 3: Ablation study on hand and object topology on HO3Dv3 dataset.

| Settings | | Metrics | |
|---|---|---|---|
| Hand Topo. | Obj. Topo. | FID | LPIPS |
| ✓ | ✓ | **41.3** | **0.171** |
| ✓ | ✗ | 47.3 | 0.183 |
| ✗ | ✓ | 47.8 | 0.187 |

**Source feature transfer.** To further enhance the refinement procedure of the coarse target hand image, we transfer multi-layers features from the source hand-object generator according to the transformation flow. In Table 4, we analyze the importance of the source feature transfer. The *None* variant refers to the framework without source feature transfer. The *Bilinear* and *Attention* variants refer to the framework with different samplers for feature transferring, *i.e.*, a bilinear sampler and an attention sampler, respectively. It can be seen that the source feature transfer provides useful cues for target image generation and the attention sampler achieves the best performance among three settings.

Table 4: Ablation study on the sampling method on HO3Dv3 dataset.

| Sampler | Metrics | |
|---|---|---|
| | FID | LPIPS |
| None | 46.5 | 0.181 |
| Bilinear | 42.5 | 0.173 |
| Attention | **41.3** | **0.171** |

**Split-and-combine generation.** Considering hand and object exhibit different properties, we involve a split-and-combine (S&C) strategy in HOGAN. In this setting, we compare HOGAN with a parameter-comparable baseline model without separately generating hand and object. In Table 5, it is observed that HOGAN outperforms its variant (without S&C) by 2.9 FID and 0.036 LPIPS gain, which demonstrates this strategy benefits the quality of generated images when involving instances with different properties.

Table 5: Ablation study on the split-and-combine strategy on HO3Dv3 dataset.

| S&C | Metrics | |
|---|---|---|
| | FID | LPIPS |
| ✗ | 44.2 | 0.207 |
| ✓ | **41.3** | **0.171** |

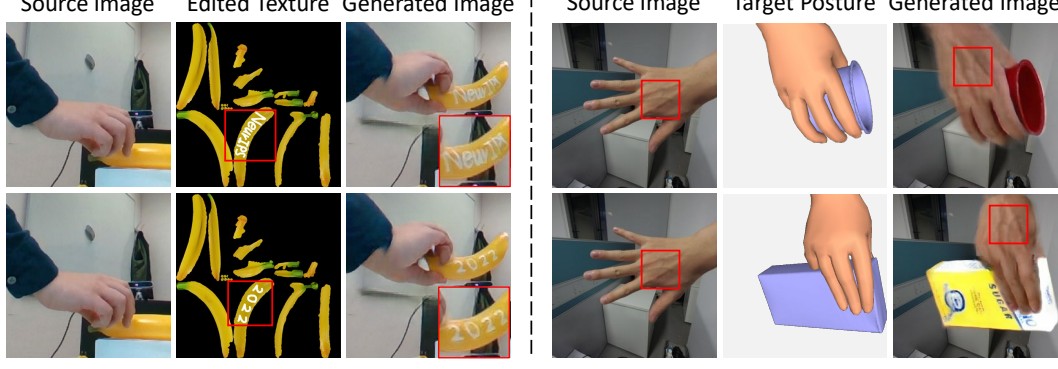

| Source Image | Edited Texture | Generated Image | Source Image | Target Posture | Generated Image |

(a) Object Texture Editing      (b) Real-World Generation

Figure 4: Applications of hand-object interaction image generation. (a) The object texture in hand-object generation is edited with characters, and HOGAN produces the images conditioned on the edited textures. (b) We test pre-trained HOGAN for the real-world hand image. As highlighted in the red box, HOGAN generates images which both maintain the hand identity in the source image and meet the target posture.

## 4.5 Application

In this subsection, we further explore several interesting applications on our HOGAN, *i.e.*, object texture editing and real-world generation. In object texture editing, we alter the object texture with characters, *i.e.*, "NeurIPS" and "2022", and generate the images conditioned on the edited textures. From Figure 4 (a), it is observed that our generated images well preserve the edited characters on the object texture. Furthermore, we take the hand image from the real scene as the source image to test our pre-trained HOGAN framework. As shown in Figure 4 (b), the generated images both maintain the source image appearance and meet the target posture condition.

These applications demonstrate the generalization of HOGAN on hands and objects for the real-world scenario and applications. For example, when a consumer is shopping online, interaction visualization will give him/her an immersive experience. Furthermore, if consumers want some customization on the object, *e.g.* adding the name on the phone, our provided application can achieve this through object texture editing. Besides, in the online shopping scenario, consumers usually do not have the object. They only need to upload a picture of their hand, and we can give them a real interaction experience via generating hand-object images with their hand identity preserved.

Based on our explored task, it is also of great importance for synthetic data creation [33]. Current HOPE methods are usually deep-learning-based, but their performances are limited by the size of training data due to the annotation cost. One way to fertilize the model is to utilize synthetic data. Our framework is also ready to generate images, which model real-world characteristics well and boost HOPE performance as shown in Table 6. Specifically, we adopt the backbone [21]. "Aug" represents the backbone is trained on both HO3D and our synthesized data. It can be observed that "Aug" outperforms the baseline (without Aug) under all metrics, especially in the object.

Table 6: Application on synthetic data creation for boosting HOPE performance.

| Aug | Hand | Object |
|---|---|---|
| | AUC | ADD-0.1D |
| ✕ | 77.2 | 67.6 |
| ✓ | **78.0** | **76.8** |

## 5 Limitations and Future Work

Our framework considers the complex spatial relationship and different appearance properties between hand and object. The effectiveness of our framework has been validated in Sec. 4. There still exist limitations in our framework. Specifically, it utilizes the dense mesh representation as the pose condition, which inevitably contains misalignment with the RGB image plane even manually annotated. This issue mainly disturbs coarse image synthesis and our framework mainly resorts to

the Hand-Object Synthesis stage for further refinement. Another way to mitigate this issue is to adopt a more effective hand-object pose estimation method for better pose representation.

Our explored HOIG task is of broad research interest to the community. When hand pose estimation turns from the isolated hand to the interacting hand-object scenario, we advocate the image generation community to draw more attention to this new HOIG task. This task involves generation of co-occurring instances under complex occlusion conditions. The advance in HOIG can also inspire other related human-centric image generation tasks. We outline the future works as follows. 1) More suitable representations can be explored to serve as a condition, jointly with their robustness analysis. 2) More applications are desirable to fertilize the HOIG task. 3) More extension to human-object interaction with the trend like PHOSA [43] can be explored.

## 6 Conclusion

In this paper, we make the *first* attempt to explore a novel task, namely hand-object interaction image generation. This task brings new challenges since it involves generating two co-occurring instances under complex interaction conditions. To deal with the challenges of this task, we propose the HOGAN framework. It explicitly considers the self- and mutual occlusion among hand and object and generates the target image in a split-and-combine manner. For comprehensive comparisons, we present baselines adopted from related single hand generation and evaluate the generated images from multiple perspectives, including fidelity and structure preservation. Extensive experiments on two datasets demonstrate our method outperforms baselines both quantitatively and qualitatively.

**Acknowledgment** This work was supported by the National Natural Science Foundation of China under Contract U20A20183 and 62021001. It was also supported by GPU cluster built by MCC Lab of Information Science and Technology Institution, USTC.

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
