# OpenReview forum: "Hand-Object Interaction Image Generation"
_NeurIPS.cc/2022/Conference — NeurIPS 2022 Accept_

### Official Review · Reviewer_p4XG · 2022-07-06

**Rating:** 4
**Confidence:** 3
**Soundness:** 2 fair
**Presentation:** 3 good
**Contribution:** 2 fair

**Summary:**

In this paper, the authors propose a new task, i.e., hand-object interaction image generation. The proposed method uses occlusion-aware topology modeling, which captures the occlusion relationship between objects and hands, and warps the unoccluded region. The warped hand input and object input, together with their topologies, are fed into the hand-object generator. The hand-object generator deals with the hand and object generation separately.

The contributions lie in that 1) the paper proposes a new task; 2) the proposed method generates plausible results for this new task.

**Questions:**

1) Apart from proposing a new task, the author should further clarify their technical contribution.
2) With the inputs containing rich prior information, the author should address the key technical challenges. In my understanding, with these inputs, the challenges mentioned in the paper are addressed. I wonder if there are any other technical challenges after using these complicated inputs. Please refer to my concerns about this issue in weakness (1).

**Limitations:**

The authors have addressed the limitations. For example, the posture representation is the dense mesh representation, which is not easy to get in real-world applications.

**Strengths And Weaknesses:**

Strengths:
1) The proposed method is intuitive and achieves plausible results on hand-object interaction generation.
2) The paper is well-written and easy to follow.


Weaknesses:
1) The proposed method takes too much information containing rich priors as inputs, which greatly eases the tasks. Specifically, with the source posture, it is easy to model the occlusion between hands and objects. With the source posture and target posture, the only thing the network needs to learn is to inpaint the warped images. As for the object generation, with the object model, the network only needs to adjust the lighting of the object. Admittedly, the task itself is challenging as suggested by the author. However, with these complicated inputs, the task is easy to solve. In my understanding, the authors tackle the problem by feeding more inputs rather than solving the problem in the technical aspect.
2) The comparison with GestureGAN+OBJ is not fair. The input to the GestureGAN is only the source image and target pose represented by sparse keypoints. The input contains much less prior information compared to the proposed method.

---

> ### Author Response · Authors · 2022-08-02
> **Author Response - Reviewer p4XG**
>
> We sincerely appreciate your valuable and constructive comments.
> Our detailed responses are listed below and we revise the manuscript accordingly.
>
> **Q1: Clarification on the necessity of current input information and challenges of leveraging the information in the adopted inputs.**
>
> **A1:** Logically, current inputs are the necessary conditions to tackle this challenging task and they can be easily obtained via the method like [1], [2].
> Furthermore, the baseline like MG2T + OBJ, which adopts similar inputs, does not produce satisfying results.
> Therefore, it is non-trivial to design a framework which can take advantage of these inputs and our technical contribution focuses on the designed methodology to tackle this task.
>
> Firstly, it is difficult to handle the complex self- and mutual occlusion during hand-object interaction.
> Linking the source 3D posture and 2D appearance under the context of two co-occurring interacting instances has not been attempted in previous generative methods.
> We achieve it efficiently through building the unified surface map to explicitly consider their occlusion conditions and provide abundant topology information inherent in model for the following target image generation.
> Leveraging these information, the hand-object generator further considers different characteristics of hand and object and generates the target image in a split-and-combine manner.
>
> [1] Shreyas Hampali et al. "Honnotate: A method for 3D annotation of hand and object poses." CVPR 2020.
>
> [2] Zhe Cao, et al. "Reconstructing hand-object interactions in the wild." ICCV 2021.
>
> **Q2: Clarification on the comparison baseline of GestureGAN.**
>
> **A2:** Since our explored new task contains no baselines, we have to modify the methods from most related single-hand generation.
> GestureGAN and MG2T are two only existing representative methods in signle-hand generation needing to compare.
> 1\) Based on GestureGAN, we clarify our modification on it in Line 208-212.
> It utilizes sparse 2D hand keypoint with inherent ambiguity and produces inferior generation results, which indicates the necessity of our adopted input.
> 2\) We also compare with MG2T + OBJ, which adopts similar inputs with ours.
> Our method achieves better performance over it, which validates the effectiveness of our designed methodology for this task.
>
>
> **Q3: Clarification on our technical contribution.**
>
> **A3:**
> 1\) Notably, one of the main contributions in our work is the first to study the new HOIG task, which contains board research and application value.
> The exploration on the new task is non-trivial, which includes the problem formulation, clarification of the task value, comprehensive baselines and evaluation methodology.
> This contribution is the cornerstone of our work and should be counted as novelty.
>
> 2\) Under the new task formulation, we propose the HOGAN framework and its technical contribution focuses on the proposed methodology to tackle the main challenges of the HOIG task.
> a) Specifically, we propose occlusion-aware topology modeling to resolve the complex self- and mutual occlusion during interaction.
> b) During the synthesis stage, we consider different characteristics of hand and object and generate the target image in a split-and-combine manner.
> Notably, utilizing the existing building block is just the tool and not claimed as our contributions.
> We are not aiming at building new building blocks in the work.
>
> 3\) For this new HOIG task, we build comprehensive baselines and multi-perspective evaluation metrics.
> Under the evaluation metrics, extensive experiments validate the effectiveness and superiority of our method over baselines both quantitatively and qualitatively.
> Our methodology is carefully designed with explicitly considering the characteristics of this new task.
> Compared with baselines, our framework generates images without blurry, texture aliasing, and false hand-object interaction, as shown in Figure 3.
>
> 4\) For this new HOIG task, we also explore plentiful applications, *i.e.*, object texture editing, real-world hand-object generation and data augmentation for HOPE (in revision).
> These applications further enrich the value of our method and proposed task.

---

> ### Author Response · Authors · 2022-08-07
> **Look forward to further discussion with reviewer p4XG**
>
> Dear reviewer p4XG:
>
> We sincerely appreciate you for the acknowledgement, the precious review time and valuable comments.
> We have updated the new revision and highlight the changes in blue in the revision.
> We hope to further discuss with you whether or not your concerns have been addressed.
> Please let us know if you still have any unclear parts of our work.
>
> Wish you a nice weekend.
>
> Best,
>
> Authors of Paper 1135

---

> ### Author Response · Authors · 2022-08-09
> **Look forward to your feedback**
>
> Dear reviewer p4XG,
>
> We would like to thank again for your time and valuable comments.
> Hopefully, our rebuttal and new submitted revision could properly address your concerns.
> We look forward to your feedback and will appreciate it if you could upgrade your score.
>
> Wish you a nice day.
>
> Best,
>
> Authors of Paper 1135

---

### Official Review · Reviewer_zW9y · 2022-07-11

**Rating:** 4
**Confidence:** 4
**Soundness:** 2 fair
**Presentation:** 3 good
**Contribution:** 2 fair

**Summary:**

This paper presents a new task of synthesizing photorealistic images of hand-object interaction. In addition, this work introduces a strong baseline method by conditioning an image-to-image translation network with hand/object topology as well as warped texture from the source image. The experiments show that the proposed approach outperforms the naive extention of existing approaches. Also the paper provides several applications such as texture editing and real hand texture transfer.

**Questions:**

Related to the weaknesses above, if AR/VR, and online shopping are the primary applications of HOIG, why not proposing the task of “human”-object interaction image synthesis instead of hand-object? I can see the immediate applications in those domains if it were full-body, but not clear with only hands unless it’s used for bootstrapping pose estimation tasks.

**Limitations:**

The paper discusses its limitations and societal impact.

**Strengths And Weaknesses:**

The paper has the following strengths:
- The paper introduces a novel task called hand-object image synthesis. As discussed in the paper, the existing literature mostly focus on scene analysis, not synthesis.
- The proposed HOGAN presents a strong baseline for this newly introduced task. The experimental results also show that existing approaches are not suitable for this novel task.
- The paper is well written and easy to follow.

On the other hand, this work has the following weaknesses:
- While the paper constantly argues that HOIG is of broad interest to the community, the listed applications and demonstration in the paper are not convincing enough. In my view, there is no immediate applications in AR/VR or online shopping with texture editing or real-hand texture transfer only inside the small cropping of hand and object images. More immediate applications of this problem would be to use synthesized data to bootstrap the training of hand-object interaction analysis (e.g., pose estimation, object localization). I would highly recommend adding experiments to show that the presented task is useful for downstream analysis tasks similarly to [Shrivastava et al. 2017].

Learning from Simulated and Unsupervised Images through Adversarial Training
AuthorsAshish Shrivastava, Tomas Pfister, Oncel Tuzel, Josh Susskind, Wenda Wang, Russ Webb
CVPR 2017

- Another issue of this work is the expected input seems unrealistic. More specifically, it remains challenging to obtain accurate hand and object poses in a single image. Noise perturbation in input poses would be great to have to assess the robustness of each approach in the presence of pose misalignment.

While the paper presents a novel task and its strong baseline method, its significant remains unclear with the current state. As I mentioned earlier, providing more convincing use cases would make the paper stronger. Thus, I would recommend resubmitting.

---

> ### Author Response · Authors · 2022-08-02
> **Author Response - Reviewer zW9y**
>
> We sincerely appreciate your valuable and constructive comments.
> Our detailed responses are listed below and we revise the manuscript accordingly.
>
> **Q1: Clarification on the application and demonstration of hand-object interaction, and difference with human-object interaction.**
>
> **A1:** Hand-object interaction is quite different from human-object interaction in multiple aspects, especially granularity (scale).
> Human-object usually focuses on more holistic interaction between the body and the object, *e.g.* bicycle, skateboard, suitcase and *etc.*
> In contrast, hand-object usually focuses on the fine-grained interaction inside the view between hand and object like phone, pencil, bottle, and *etc.*
> This task is also of great importance and contains application scenarios like AR/VR and online shopping.
>
> For example, when a consumer is shopping online, interaction visualization will give him/her an immersive experience.
> Furthermore, if consumers want some customization on the object, *e.g.* adding the name on the phone, our provided application can achieve this through object texture editing, as shown in Figure 4(a).
> Besides, in the online shopping scenario, consumers usually do not have the object.
> They only need to upload a picture of their hand, and we can give them a real interaction experience via generating hand-object images with their hand identity preserved, as shown in Figure 4(b).
>
>
> **Q2: More application on utilizing synthesized data to bootstrap the training of hand-object interaction analysis.**
>
> **A2:** We have cited your mentioned paper and added another application, *i.e.,* utilizing synthesized data to boost the performance of hand-object pose estimation.
> As shown in Table 1, we adopt the backbone in [21].
> "Baseline" denotes directly training the backbone on HO3D dataset, while "Baseline + Aug" represents the method training on both HO3D and our synthesized data.
> It can be observed that "Baseline + Aug" outperforms the baseline method under all metrics, especially in the object pose.
>
> Table 1. Application on bootstrapping the training of hand-object interaction analysis.
>
> | Method            | Hand AUC   | Object Avg\. ADD\-0\.1D   |
> |:-----------------:|:----------:|:-------------------------:|
> | Baseline          | 77\.2      | 67\.6                     |
> | Baseline \+ Aug   | **78\.0**  | **76\.8**                 |
>
> **Q3: Clarification on the input issue and robustness analysis.**
>
> **A3:** The expected input of our framework is easy to obtain automatically through the method like [1].
> To demonstrate the robustness of each approach, we add noise on the input pose of each method to mimic the case of pose misalignment and evaluate its impact on the generation quality.
> In practice, we perturb the pose by adding random noise with the range of 30\% of its magnitude.
> As shown in Table 2, our method demonstrates robustness on pose misalignment, *i.e.*, only +0.6 FID and -0.004 LPIPS due to perturbation, which exhibits more robustness than MG2T + OBJ.
> GestureGAN + OBJ still produces very blurry results.
> Although the perturbation affects its generated images, the performance keep relatively unchanged.
> Compared with them, our method still achieves the best performance after noise perturbation.
>
> [1] Shreyas Hampali et al. "Honnotate: A method for 3D annotation of hand and object poses." CVPR 2020.
>
> Table 2. Comparison with baselines on pose information w/ and w/o perturbation.
>
> | Method | w/o | perturbation | w/ | perturbation |
> |:------:|:---:|:------------:|:--:|:------------:|
> |        | FID |  LPIPS  | FID | LPIPS |
> | GestureGAN + OBJ | 82\.0 | 0\.316 | 82\.1 | 0\.316 |
> | MG2T + OBJ | 45\.6 | 0\.214 | 48\.9 | 0\.219 |
> | HOGAN | **41\.9** | **0\.172** | **42\.5** | **0\.176**|

---

> ### Author Response · Authors · 2022-08-07
> **Look forward to further discussion with reviewer zW9y**
>
> Dear reviewer zW9y:
>
> We sincerely appreciate you for the precious review time and valuable comments.
> We have provided corresponding responses, experiment results on bootstrapping the training of hand-object interaction analysis, and updated the new revision, which we believe have covered your concerns.
> For convenience, we highlight the changes in blue in the revision.
> We hope to further discuss with you whether or not your concerns have been addressed.
> Please let us know if you still have any unclear parts of our work.
>
> Wish you a nice weekend.
>
> Best,
>
> Authors of Paper 1135

---

> ### Author Response · Authors · 2022-08-09
> **Look forward to your feedback**
>
> Dear reviewer zW9y,
>
> We would like to thank again for your time and valuable comments.
> Hopefully, our rebuttal and new submitted revision could properly address your concerns.
> We look forward to your feedback and will appreciate it if you could upgrade your score.
>
> Wish you a nice day.
>
> Best,
>
> Authors of Paper 1135

---

### Official Review · Reviewer_aYmL · 2022-07-11

**Rating:** 10
**Confidence:** 5
**Soundness:** 4 excellent
**Presentation:** 4 excellent
**Contribution:** 4 excellent

**Summary:**

The authors have investigated the task of hand-object interaction generation (HOIG) which is inverse problem of hand-object pose estimation (HOPE) problem.

Target image for hand, object, and the interaction between the two is created using a split-and-combine manner.

HOGAN architecture based on conditional GANs is proposed for creating a unified space for hand and object. In their task, they have synthesized hand and object interaction conditioned on a target pose while preserving the appearance of the source image. This framework consists of background, object, and hand streams. Eventually, using a fusion model, all these streams are merged.

The hand-object interaction generation framework is finally tested on two large-scale datasets, HO3Dv3 and DexYCB that have annotated hand-object mesh representations.

The generated interaction hands and objects are finally evaluated both quantitatively and qualitatively under various methods for fidelity (e.g. FID), structure preservation (e.g. PA-MPJPE), as well as human subject evaluations.


**Questions:**

1- How can your model apply to non-rigid object interaction?

2- How would you expand your model to hand-object interaction without an initial source?

3- How does your model perform in case of non-symmetric objects vs symmetric objects?

4- Does your model perform greatly independently of the object size? For example, does it do a great job when interacting with a soccer ball rather than holding a toothbrush?

5- In figure 2, Inpainted Background, why didn’t you remove the entire hand? Wouldn’t the remaining of hand cause you problem further down the line?

6- Line 180: please state how you choose your hyperpatameters such as LR, number of epochs, etc. Did you perform any hyperparameter tuning (e.g. Grid Search or Random Search)? Please provide the mechanism for so as well as potential results of your hyperparameter tuning.

7- Have you tested your model with darker color hands? How does your model perform in such cases? Is your framework agnostic to hand texture, size, and color? For example, if someone is wearing a ring on her finger, how does your model handle the reconstruction of the ring in the target pose?


Suggestions:

1- VR / AR → AR/VR

Line 68: face reenactment and sign language production → face reenactment, and sign language production

Line 171: please state which VGG, e.g. VGG-16 or VGG-19 in the writing

Line 175: implementation details and evaluation metrics → implementation details, and evaluation metrics



**Limitations:**

It is stated that the link code is provided but I cannot find it in the paper. Please provide the link to the source-code. It is also not clear to me if the code to modified baselines as well as HOGAN framework would be open-sourced in case the paper gets published.

I do see the code in the supplementary materials however it is just not clear to me if the code will be released in an open-source fashion. Also, please let us if you would share the trained model checkpoints for other researchers publicly.




**Strengths And Weaknesses:**

This research work is of great importance because its direct use in AR/VR applications such as augmented shopping or mixed-reality gaming experiences.

Their framework processes complex self- and mutual occlusion for both hand and object. For example, because hand is articulated, there is self-occlusion between the hand joints.
Based on their claims, these authors are the first to study the HOIG task which counts as novelty. Their other aspect of novelty is propose of HOGAN architecture for conditional generation of interacting hands and objects.

I think it’s very impressive that the correct texture is rendered for the object in different pose despite the inherent occlusion.

The comprehensive benchmarking and evaluation methodology followed by ablation studies by author is a very strong point in this paper. Further, given that the authors are the first to define this new task and there’s a lack of baselines, they did a great job of modifying the closest baselines to their needs and use them for comparison.

Not really weakness, but adding a paragraph or two about importance of synthetic dataset (not just HOI dataset) creation and its application in deep learning would be to the benefit of this paper.

Weakness: Provide mechanism for hyperparamater tuning of your framework hyperparamaters. Question 3.b answer states the author have told how they have chosen the hyperparameters but it is not provided.

I think there might have been a need for IRB due to use of human subjects for qualitative analysis of produced HOI. However, in Question 5.b, the answer is N/A. I would say the qualitative study should be outsourced to others not the authors because if it is done by the authors, there is a lot of bias involved. Please provide some clearance on this.

It is not clear to me how the participants for the qualitative analysis are hired. Are they hired in-person or through an online crowdsourcing platform such as Amazon Turk? In either case, please provide details of subject recruiting.

---

> ### Author Response · Authors · 2022-08-02
> **Author Response - Reviewer aYmL - Part 1**
>
> We sincerely appreciate your valuable and constructive comments.
> Our detailed responses are listed below and we revise the manuscript accordingly.
>
> **Q1: Importance of synthetic dataset creation and its application in deep learning.**
>
> **A1:** Based on our explored task, it is also of great importance for synthetic data creation, which contains the potential application to boost the performance on hand-object interaction pose estimation (HOPE).
> Current HOPE methods are usually deep-learning-based, but their performances are limited by the size of training data due to the annotation cost.
> One way to fertilize the model is to utilize synthetic data.
> Our framework is also ready to generate images, which model real-world characteristics well and boost HOPE performance.
> We have verified the effectiveness of synthetic dataset creation in the aspect of boosting HOPE performance as shown in Table 1.
> As shown in Table 1, we adopt the backbone in [21].
> "Baseline" denotes directly training the backbone on HO3D dataset, while "Baseline + Aug" represents the method training on both HO3D and our synthesized data.
> It can be observed that "Baseline + Aug" outperforms the baseline method under all metrics, especially in object localization.
>
> Table. 1 Application on bootstrapping the training of hand-object interaction analysis.
>
> | Method            | Hand AUC   | Object Avg\. ADD\-0\.1D   |
> |:-----------------:|:----------:|:-------------------------:|
> | Baseline          | 77\.2      | 67\.6                     |
> | Baseline \+ Aug   | **78\.0**  | **76\.8**                 |
>
>
> **Q2: Mechanism for hyper-parameter tuning.**
>
> **A2:** We choose the hyper-parameter by our experience.
> For the number of epochs, we keep it consistent among all methods for fair comparison.
> Grid search may bring better performance.
>
>
> **Q3: Clarification on the details of qualitative analysis.**
>
> **A3:** Our IRB application has been granted by our institution.
> Due to the double-blind reviewing policy, we will release the approval once our paper gets accepted.
> The qualitative study is outsourced to the hired participants, without any author of this paper.
> We hire the participants in person and we randomly hire the participants in our college.
> We have estimated the hourly wage in our local region and have paid them a gift of equal value.
>
>
> **Q4: How the model applies to non-rigid object interaction.**
>
> **A4:** Since the available object models are rigid, it is hard to model the non-rigid object interaction in the current form.
> As future work, if we can get access to the object model with the non-rigid modeling capability, which contains the parameters depicting the non-rigid deformation, we can further explore our model to be applied in this scenario.
>
>
> **Q5: How to expand the model to hand-object interaction without an initial source.**
>
> **A5:** The initial source is an important information for our framework, which provides the appearance of hand for the generated images.
> A possible approach to expand the model without an initial source is to use pre-defined hand surface as texture information to generate images.
> However, this extension may need a higher cost, since it needs to collect the realistic hand texture.
>
>
> **Q6: How the model performs in case of non-symmetric objects vs symmetric objects.**
>
> **A6:** Our model does not contains the assumption on the symmetric characteristic of the object.
> Therefore, our model performs fairly well regardless of whether the object is symmetric.
> We visualize some samples in [here](https://anonymous.4open.science/r/HOIG-2F04/assets/symmetric.png).
>
> **Q7: Whether the model performs greatly independently of the object size.**
>
> **A7:** Since our model does not contain the assumption on the object size, our model performs consistently well independently of the object size.
> The drill and banana exhibit large differences in object size and appearance.
> As shown in Figure 3, the hand-drill and hand-banana images generated by our model are consistent with their ground-truth images.
>
> **Q8: Clarification on the details on the inpainted background branch.**
>
> **A8:** Since MANO only models the hand without the arm and there are no indicators on the arm, it is hard to remove the entire arm part in the inpainted background.
> We will leave it as future work.

---

> > ### Author Response · Authors · 2022-08-02
> > **Author Response - Reviewer aYmL - Part 2**
> >
> > **Q9: Results with darker color hands and whether the framework is agnostic to hand texture, size, and color.**
> >
> > **A9:** We have tested the darker color hand and the images are shown in the [here](https://anonymous.4open.science/r/HOIG-2F04/assets/dark_hand.png).
> > It can be observed that the generated images still exhibit consistent structure with the target posture and preserve the identity in the source image.
> > Our framework is agnostic to the hand texture, size, and color.
> > We test our framework on the real-world hand image in Figure 4 (b).
> > As shown in the highlighted red box, the generated image maintains the hand identity well.
> > However, MANO only models the hand itself, without modeling the ring attached to it and we are not able to access the whole ring information via the source image.
> > Therefore, if the finger wears the ring, it is hard to reconstruct the ring in the target pose.
> >
> > **Q10: Response to the "Suggestions" section.**
> >
> > **A10:** We will correct them following your suggestions. The VGG is implemented as VGG-19.
> >
> > **Q11: Clarification on the following open source.**
> >
> > **A11:** Due to the double-blind reviewing policy, current source code is shown in the anonymously open-sourced Github with the link [https://anonymous.4open.science/r/HOIG-2F04/](https://anonymous.4open.science/r/HOIG-2F04/).
> > We will release our modified baselines, our proposed framework, and their pre-trained model checkpoints publicly to Github once our paper gets accepted.

---

### Official Review · Reviewer_Bc9p · 2022-07-11

**Rating:** 4
**Confidence:** 4
**Soundness:** 3 good
**Presentation:** 3 good
**Contribution:** 2 fair

**Summary:**

The paper presents a framework to synthesize images with hand-object interactions using the 3D model of the object and the hand pose (also in 3D). The paper also claims to introduce a new task named hand-object interaction image generation. A set of experiments in two different datasets showed that the proposed framework outperformed constructed baselines in several metrics, including LPIPS, FID, AUC, PA-MPJPE, and ADD-01.D.

**Questions:**

1. Were all UNet trained from scratch or pre-trained using some dataset in an inpaint task?
2. It is not clear the fusion step. Is it a learning layer? If so, is it an MLP?
3. Is the hand-object generator training end-to-end? Are all the networks being training simultaneously, or was used a training-wise strategy?
4. There are three independent branches in the network; therefore, it seems that each branch is unaware of the other, and the interaction between the hand and the object is not considered when training the model. How does the framework manage to explore the hand-object interaction to generate a more realistic image?
5. At last, following my concern about the complexity and necessity of using a learning approach to solve the problem, why would not a good solution apply a rendering algorithm and after improving the photorealism of the synthetic image using a style transfer method? This kind of solution feels more practical and less complex than the proposed framework.


**Limitations:**

Yes. The limitations were adequately addressed in the Limitations and Future Work section.

**Strengths And Weaknesses:**

- Strengths:
    - The paper is well-presented and easy to understand. No obvious typos.
    - The technical novelty looks incremental, but the experimental results and conclusions may be interesting for the community.
    - The proposed framework is well-validated, and the results showed clearly that it is superior to the baselines.

- Weakness:
    - There are two major weaknesses. First, it seems that the proposed framework is too complex to solve the problem. Since it is provided a 3D model for the object and the hand indicating the pose of both, why not apply a photorealistic render algorithm and compose the image using the background of the input image?
    - The second weakness is the novelty of the technical solution feels incremental. Most of the building blocks of the proposed frameworks come from related work (e.g., UNets and SPADE). For instance, after computing a projection of the hand with the missing texture (occlusion-aware topology modeling component), the hand-object generator is basically applying a UNet to inpaint the hands and object texture and using SPADE to take into account the three-dimensional shape.
    - Another concern refers to the baselines and the evaluation metrics, which seem they are not challenging the proposed framework. For example, why not compare against a photorealist render algorithm and not use a pixel-wise error (e.g., MSE) using the ground-truth images to evaluate the quality of image generation?

---

> ### Author Response · Authors · 2022-08-02
> **Author Response - Reviewer Bc9p - Part 1**
>
> We sincerely appreciate your valuable and constructive comments.
> Our detailed responses are listed below and we revise the manuscript accordingly.
>
>
> **Q1: Clarification on utilizing the photo-realistic rendering-related algorithm to solve this problem and comparison with the photo-realistic rendering algorithm.**
>
> **A1:**
> 1\) For photo-realistic rendering, one main procedure is to get the reliable texture.
> However, the source image only contains the partial hand texture.
> That is to say, if we extract the hand texture from the source image and render the hand-object image under the guidance of the target posture, the hand part of the generated image is incomplete.
> Therefore, the direct rendering method is not applicable.
>
> 2\) As suggested, after applying a rendering algorithm, we can further improve the photo-realism of the synthetic image using a style transfer method.
> Actually, our framework has integrated this idea.
> In the "occlusion-aware topology modeling" stage, the extracted visible hand texture in the source image and pre-stored object texture are rendered to the target image plane as the coarse disentangled hand-object image ("Hand Input" and "Object Input"), as shown in the lower-left corner of Figure 2.
> During the "hand-object generator" stage, these coarse images are further refined for the target hand-object image with their different characteristics considered.
>
> 3\) We have compared with a photo-realistic rendering-related method, *i.e.*, MG2T + OBJ.
> In this method, the rendered hand and object images are further refined to produce the hand-object translation result.
> The method details are stated in Line 213-216.
> Our method outperforms it under all metrics, which demonstrates the necessity and effectiveness of our framework design.
>
>
> **Q2: Clarification on the technical contribution.**
>
> **A2:**
> 1\) Notably, one of the main contributions in our work is the first to study the new HOIG task, which contains board research and application value.
> The exploration on the new task is non-trivial, which includes the problem formulation, clarification of the task value, comprehensive baselines and evaluation methodology.
> This contribution is the cornerstone of our work and should be counted as novelty.
>
> 2\) Under the new task formulation, we propose the HOGAN framework and its technical contribution focuses on the proposed methodology to tackle the main challenges of the HOIG task.
> a) Specifically, we propose occlusion-aware topology modeling to resolve the complex self- and mutual occlusion during interaction.
> b) During the synthesis stage, we consider different characteristics of hand and object and generate the target image in a split-and-combine manner.
> Notably, utilizing the existing building block is just the tool and not claimed as our contributions.
> We are not aiming at building new building blocks in the work.
>
> 3\) For this new HOIG task, we build comprehensive baselines and multi-perspective evaluation metrics.
> Under the evaluation metrics, extensive experiments validate the effectiveness and superiority of our method over baselines both quantitatively and qualitatively.
> Our methodology is carefully designed with explicitly considering the characteristics of this new task.
> Compared with baselines, our framework generates images without blurry, texture aliasing, and false hand-object interaction, as shown in Figure 3.
>
> 4\) For this new HOIG task, we also explore plentiful applications, *i.e.*, object texture editing, real-world hand-object generation and data augmentation for HOPE (in revision).
> These applications further enrich the value of our method and proposed task.
>
>
>
> **Q3: Clarification on not using pixel-wise error as the evaluation metric.**
>
> **A3:** The MSE evaluation metric usually prefers the blurry result and is currently less utilized in generative problems.
> In this work, we resort to the most widely-used metrics, *i.e.*, FID and LPIPS, in generative problems for the fidelity of the generated image.
> FID measures the distances between the real-image distribution and generated-image distribution.
> LPIPS is a weighted perceptual similarity between the generated image and the ground-truth image, which matches the human perception.
> These two metrics are commonly adopted for evaluation in [1, 2, 3].
>
> [1] Caroline Chan *et al.* "Everybody Dance Now." ICCV 2019.
>
> [2] Tero Karras *et al.* "Analyzing and Improving the Image Quality of StyleGAN." CVPR 2020.
>
> [3] Fabian Mentzer *et al.* "High-Fidelity Generative Image Compression." NeurIPS 2020.
>
> **Q4: Clarification on U-Net.**
>
> **A4:** All U-Nets are trained from scratch.
>
>
> **Q5: Clarification on the fusion step.**
>
> **A5:** As mentioned in Line 152 - 159, the fusion step is implemented via learning the hand-object mask and the hand mask by two convolutional layers and fusing the three-stream results as Equation 7.

---

> > ### Author Response · Authors · 2022-08-02
> > **Author Response - Reviewer Bc9p - Part 2**
> >
> > **Q6: Clarification on the training of the hand-object generator.**
> >
> > **A6:**  The hand-object generator is trained in an end-to-end manner and all the networks are being trained simultaneously.
> >
> > **Q7: Clarification on the framework exploring the hand-object interaction.**
> >
> > **A7:** In the "occlusion-aware topology modeling" stage, we explicitly explore hand-object interaction.
> > Specifically, we build the unified surface space, and consider the complex self- and mutual occlusion during hand-object interaction.
> > With this stage, the complex relationship between hand and object is disentangled and the visible parts of hand and object are synthesized aligning the target image plane, respectively.
> > Considering the different characteristics of hand and object, the hand-object generator further produces the final target image in a split-and-combine manner.

---

> ### Author Response · Authors · 2022-08-07
> **Look forward to further discussion with reviewer Bc9p**
>
> Dear reviewer Bc9p:
>
> We sincerely appreciate you for the precious review time and valuable comments.
> We have provided corresponding responses and updated the new revision, which we believe have covered your concerns.
> For convenience, we highlight the changes in blue in the revision.
> We hope to further discuss with you whether or not your concerns have been addressed.
> Please let us know if you still have any unclear parts of our work.
>
> Wish you a nice weekend.
>
> Best,
>
> Authors of Paper 1135

---

> ### Author Response · Authors · 2022-08-09
> **Look forward to your feedback**
>
> Dear reviewer Bc9p,
>
> We would like to thank again for your time and valuable comments.
> Hopefully, our rebuttal and new submitted revision could properly address your concerns.
> We look forward to your feedback and will appreciate it if you could upgrade your score.
>
> Wish you a nice day.
>
> Best,
>
> Authors of Paper 1135

---

### Comment · Area_Chair_TTZP · 2022-08-08
**Any thoughts from reviewers**

Hi Reviewers,

The discussion period is closing soon. Please take a look at the responses from the authors. If you have further questions, please ask them now, since the authors will be unable to respond soon. It's substantially more productive, effective, and reasonable to have a quick back-and-forth with authors now than to raise additional questions or concerns post-discussion period that the authors are unable to address.

Thanks,

AC

---

### Meta-Review · Area_Chair_TTZP · 2022-08-24

**Recommendation:** Accept
**Confidence:** Certain

**Metareview:**

On the surface, this paper seems to be split between three borderline rejects (4) and one strong champion of the paper (10). However, this is not the full story, since two of the reject-inclined reviewers, Bc9p and zW9y did not participate post-rebuttal, despite multiple prods from the AC. The AC examined the stated weaknesses from Bc9p and zW9y, the authors' response to them, as well as the paper. The AC does not think that the concerns are paper stopping if they were left unaddressed (e.g., a glaring experimental weakness, an incorrect statement) and moreover finds the authors' response to these concerns satisfactory. The AC is then left with the reviews from p4XG (4) and aYmL (10). The remaining primary concern comes from p4XG, who points out that the method has a lot of inputs, which makes the problem considerably easier. The AC understands this concern (and thinks lowering the input requirements would be a great next step), but on balance is inclined to accept the paper. This is motivated by the quality of the results as well as aYmL's enthusiasm for the work. The AC would encourage the authors to use the extra page to give clear responses to the reviewers' questions  (e.g., what are the applications, why not just do a direct render) in the final version of the paper. Others will have similar questions.


**Award:**

No

---

### Decision · Program_Chairs · 2022-09-14

Accept